# Error Correction Output Codes for Robust Neural Networks against Weight-errors: A Neural Tangent Kernel Point of View

**Anlan Yu**[*]
Lehigh University
any218@lehigh.edu

**Shusen Jing**[*]
University of California, San Francisco
shusen.jing@ucsf.edu

**Ning Lyu**
Lehigh University
nil418@lehigh.edu

**Wujie Wen**
North Carolina State University
wwen2@ncsu.edu

**Zhiyuan Yan**
Lehigh University
zhy6@lehigh.edu

## Abstract

Error correcting output code (ECOC) is a classic method that encodes binary classifiers to tackle the multi-class classification problem in decision trees and neural networks. Among ECOCs, the one-hot code has become the default choice in modern deep neural networks (DNNs) due to its simplicity in decision making. However, it suffers from a significant limitation in its ability to achieve high robust accuracy, particularly in the presence of weight-errors. While recent studies have experimentally demonstrated that the non-one-hot ECOCs with multi-bits error correction ability, could be a better solution, there is a notable absence of theoretical foundations that can elucidate the relationship between codeword design, weight-error magnitude, and network characteristics, so as to provide robustness guarantees. This work is positioned to bridge this gap through the lens of neural tangent kernel (NTK). We have two important theoretical findings: 1) In clean models (without weight-errors), utilizing one-hot code and non-one-hot ECOC is akin to altering decoding metrics from $l_2$ distance to Mahalanobis distance. 2) There exists a threshold, determined by the normalized distance among codewords, the DNN architecture, and the scale of weight-errors. If the distance between a clean output (without weight-errors) and its nearest codewords is smaller than this threshold, then the DNN can make predictions as if it is free of weight-errors. Based on these findings, we further demonstrate how to practically use them to identify optimal ECOCs for simple tasks (small number of classes) and complex tasks (large number of classes), by balancing the code orthogonality (as per finding 1) and code distance (as per finding 2). Extensive experimental results across four datasets and four DNN models validate the superior performance of constructed codes, guided by our findings, compared to existing ECOCs. To our best knowledge, this is the first work providing theoretical explanations for the effectiveness of ECOCs and offers associated design guidance for optimal ECOCs specifically tailored to DNNs.

## 1 Introduction

Inspired by error correction codes in wireless communication and memory system, error correction output codes (ECOCs) are proposed to improve the generalization performance of multi-class classification in decision trees by decomposing a complex problem into simpler binary classification

---

[*]These authors contributed equally to this work

38th Conference on Neural Information Processing Systems (NeurIPS 2024).

tasks [1]. As deep neural networks (DNNs) are ever-increasingly popular in machine learning, one-hot code, as a special ECOC providing one bit protection, has been most widely adopted in the output layer of modern DNNs due to its simplicity in decision making and decent generalization performance. However, it experiences notable accuracy degradation in the presence of weight errors, such as those originating from hardware defects in non-volatile compute-in-memory (NVCiM) DNN accelerators [2, 3, 4, 5, 6, 7].

A number of works has adopted general ECOCs (non-one-hot code) to improve DNN's robust accuracy against weight errors [8, 9, 10, 11, 12]. While these studies demonstrate that ECOCs with stronger error correction capability, can outperform one-hot codes to some extent through experiments, they often directly apply existing ECOCs to DNNs, with no explicit code design optimization tailored to modern DNNs. In this regard, there lacks a systematic theoretical study to answer the following key questions: *1) What is the mechanism behind ECOC's efficacy in DNNs? 2) How effective can ECOC be? 3) How optimal ECOCs tailored for DNNs can be designed principally?* To bridge this gap and provide insights to code construction, in this work, we overcome the challenges of characterizing the connection between the actual DNN performance and the application of ECOCs, and answer these questions rigorously based on novel theoretical proof. For the first question, the prevailing intuition in existing research, which suggests that enlarging the distance among codewords enhances robustness, is somewhat imprecise. In fact, we prove that the normalized distance (by the square root of the code length) among codewords is the crucial quantity in improving the DNNs' robustness. For the second question, we establish that there exists a threshold, determined by the normalized distance among codewords, DNN architecture, and the scale of weight-errors. If the distance between a clean output (in the absence of weight-errors) and its nearest codewords is smaller than this threshold, the DNN can make predictions as if it is free of weight-errors. For the third question, we analyze the overall performance of ECOCs by separating it into clean performance (in the absence of weight-errors) and performance degradation due to weight-errors. We demonstrate that the clean performance is influenced by the correlation among ECOC codewords (code orthogonality), while robustness is determined by the normalized distance among the codewords (code distance). Thus, both aspects should be carefully considered during code construction.

**Contributions:** To this end, we provide a theoretical characterization of the efficacy of ECOCs on DNNs through the lens of the neural tangent kernel (NTK). Building on our theoretical insights, we propose two ECOC construction methods tailored to DNNs focusing on small tasks and complex tasks. Our contributions are summarized as follows:

• In clean models (without weight errors), utilizing one-hot code and non-one-hot ECOC is akin to altering decoding metrics from $l_2$ distance to Mahalanobis distance.

• We prove that there exists a threshold, determined by the normalized distance among codewords, the DNN architecture, and the scale of weight-errors. If the distance between a clean output (in the absence of weight-errors) and its nearest codewords is smaller than this threshold, the DNN can make predictions as if it is free of weight-errors.

• Inspired by our theoretical results, we propose two ECOC construction methods optimizing the trade-off between codewords orthogonality and distance among codewords.

• Extensive experimental results on four datasets and four DNN models show that our constructed codes, based on our findings, surpass the performance of existing ECOCs with up to $7\%$.

To the best of our knowledge, this is the first work providing theoretical explanations for the effectiveness of ECOCs and offers associated design guidance for optimal ECOCs specifically tailored to DNNs.

## 2 Related Works

**Error Correction Output Codes** Previous works have adopted ECOCs to improve the robustness of DNNs [8, 9, 10, 12, 11]. Gupta *et al.* construct ECOCs by maximizing row-wise and column-wise Hamming distances together [12]. They particularly focus on short codes and weight-error free scenarios, instead of NN's robustness against weight-errors. Yu *et al.* propose a DNN output error decorrelation framework to enhance performance of ECOCs [11]. Deng *et al.* apply ECOCs to DNNs to improve reliability and false rejection rate [8]. Liu *et al.* adopt Hamming codes to improve the robustness of DNNs [9]. Verma *et al.* claim that the robustness provided by ECOCs may stem from

sigmoid activation functions, which allows larger error margin than softmax [10]. Existing works are mostly empirical studies and fail to address the theoretical foundation of ECOCs.

**Neural Tangent Kernel** Jacot *et al.* propose the concept of neural tangent kernel. They prove that the learning dynamic of an infinite-width DNN with proper initialization is essentially a dynamic of kernel ridge regression with defined neural tangent kernel [13]. Lee *et al.* refine the results in [13] and prove that infinite-width DNNs evolve like a linear model under gradient descent [14]. Lee *et al.* demonstrate that infinitely wide deep neural networks with specific types of initialization and non-linear activation functions converge to Gaussian processes [15]. These works focus on learning dynamic of the DNN in NTK regime without considering weight-errors.

## 3 Preliminaries

### 3.1 Error Correction Output Codes (ECOCs)

ECOC is a generalization of conventional one-hot codes with various code length and arbitrary binary entries. An ECOC is defined by an encoding function $\mathcal{E} : [C] \to \{1, -1\}^{n_L}$, where $C$ is the number of classes, and $[C]$ denotes the set $[C] \triangleq \{1, 2, ..., C\}$. Here $n_L$ is the code length, which is equal to the dimension of the DNN's outputs. The encoding function essentially maps a class label to a binary codeword. Let $\mathcal{D}$ denote the training data set with each entry as a pair $(x, c)$, where $x \in \mathbb{R}^{n_0}$ and $c \in [C]$ are DNN inputs and the corresponding label, respectively; $n_0$ is the input dimension of the DNN. Let $f(x; \theta) : \mathbb{R}^{n_0} \to \mathbb{R}^{n_L}$ be the DNN model parameterized with weights $\theta$. The objective during the training process is to minimize the loss function

$$\mathcal{L}(\theta) \triangleq \frac{1}{|\mathcal{D}|} \sum_{(x,c) \in \mathcal{D}} g(f(x; \theta), \mathcal{E}(c)), \tag{1}$$

where $g(\cdot, \cdot)$ is referred to as sample loss. During the inference, the DNN outputs $f(x; \theta)$ are mapped to classification decisions through the following decoding process

$$D(f(x; \theta)) \triangleq \arg \min_{c \in [C]} \|f(x; \theta) - \mathcal{E}(c)\|, \tag{2}$$

where $D(\cdot) : \mathbb{R}^K \to [C]$ is the decoding function. We can choose arbitrary metric in Eq. (2) for decoding, but we mainly focus on the $l_2$ norm in this paper.

### 3.2 Neural Tangent Kernel (NTK)

NTK is a tool for analyzing the learning dynamic and generalization of DNNs. It allows us to study DNNs in a reproducible kernel Hilbert space (RKHS) instead of intractable DNN weights space. We adopt a fully-connected feed-forward neural networks with $L$ layers. At the $(l + 1)$-th layer, we have

$$\begin{cases} h_{l+1} = w_{l+1} x_l + b_{l+1} \\ x_{l+1} = \phi(h_{l+1}) \end{cases} \tag{3}$$

for $l = 0, 1, ..., L - 2$, where $h_{l+1} \in \mathbb{R}^{n_{l+1}}$ and $x_{l+1} \in \mathbb{R}^{n_{l+1}}$ are the pre- and post-activation values at the $(l + 1)$-th layer; $w_{l+1} \in \mathbb{R}^{n_{l+1} \times n_l}$ and $b_{l+1} \in \mathbb{R}^{n_{l+1}}$ are the weights and bias accordingly; $n_l$ is the width of the $l$-th layer; $\phi(\cdot)$ is the activation function. In this case, $n_0$ is the dimension of DNN inputs and $n_L$ is the dimension of DNN outputs (also the ECOC code length). With NTK parameterization, the entries of $w_{l+1}$ and $b_{l+1}$ are initialized with independent and identically distributed (i.i.d.) Gaussian random variables with $\mathcal{N}(0, \frac{1}{n_l})$ and $\mathcal{N}(0, 1)$, respectively.

Let $\mathcal{X} \in \mathbb{R}^{n_0 \times |\mathcal{D}|}$ be the input set in $\mathcal{D}$, where each column represents a sample of inputs, and let $\mathcal{Y} \in \mathbb{R}^{n_L \times |\mathcal{D}|}$ be the corresponding codewords, where each column represents the corresponding target. Without loss of generality, we parameterize $n_l = \alpha_l n$ with some $\alpha_l > 0$ and $n > 0$, for $1 \le l \le L - 1$, then training DNNs in the NTK regime minimizing MSE will result in the following DNN function according to [13] when hidden layer width parameter $n \to \infty$,

$$f(x) = \mathcal{Y} \mathcal{K}(\mathcal{X}, \mathcal{X})^{-1} \mathcal{K}(\mathcal{X}, x) \tag{4}$$

where $\mathcal{K} : \mathbb{R}^{n_0} \times \mathbb{R}^{n_0} \to \mathbb{R}$ is the NTK, $\mathcal{K}(\mathcal{X}, \mathcal{X}) \in \mathbb{R}^{|\mathcal{D}| \times |\mathcal{D}|}$, and $\mathcal{K}(\mathcal{X}, x) \in \mathbb{R}^{|\mathcal{D}| \times 1}$.

# 4 Theoretical Results

In this section, we analyze the efficacy of ECOCs on DNNs through the lens of NTK. To the best of our knowledge, this represents the first theoretical exploration of ECOCs' effectiveness on DNNs. Our findings are summarized as follows:

• We prove that adopting ECOC as a replacement of one-hot codes is equivalent to changing the decoding metric from $l_2$ distance to a Mahalanobis distance.

• We establish an upper bound for the perturbation of DNN outputs, suggesting that the normalized distance of code is a crucial factor determining the error correction capability of ECOCs.

## 4.1 Assumptions

**Assumption 1.** *The activation functions in the hidden layers are bounded at 0, i.e., $\phi(0) < \infty$, and they are Lipschitz continuous with parameter $B$, i.e., for any $h$ and $h'$, we have*

$$|\phi(h) - \phi(h')| \leq B|h - h'|. \tag{5}$$

**Assumption 2.** *For any DNN input $x_0$, we have $\frac{\|x_0\|_2}{\sqrt{n_0}} \leq 1$.*

**Assumption 3.** *The matrix $\mathcal{K}(\mathcal{X}, \mathcal{X}) \in \mathbb{R}^{|\mathcal{D}| \times |\mathcal{D}|}$ is full-rank.*

**Assumption 4.** *The initial gradient with respect to the DNN output is bounded, i.e., there exists a constant $R_0$ such that $\left\|\nabla_{f(\mathcal{X}, \theta_0)} \mathcal{L}\right\|_2 < R_0$.*

Assumption 1 holds for most of the common state-of-the-art (SOTA) activation functions, such as ReLU, sigmoid and tanh. Assumption 2 essentially assumes a bounded norm of all DNN inputs. Without loss of generality, this assumption holds after input normalization. Assumption 3 holds almost surely for smooth kernel and continuous probability distribution of input samples. The assumption here is used to avoid the corner cases of uninvertible $\mathcal{K}(\mathcal{X}, \mathcal{X})$. Assumption 4 should hold with high probability with proper choice of $R_0$ due to random initialization.

## 4.2 Efficacy of ECOCs in Absence of Weight-errors

In this subsection, we discuss the behavior of DNNs after being trained with ECOC. Our first result is based on eq. (4) proved in [13].

**Proposition 1.** *Let Assumption 1, 2, 3 and 4 hold. Suppose there is no weight-errors and hidden layer width parameter $n \to \infty$, then applying ECOC is equivalent to changing the decoding function of one-hot codes (with 0 and 1 entries) to the following*

$$D(f(x)) = \arg \min_{c \in [C]} \|f(x) - e_c\|^2_{\mathcal{E}([C])^T \mathcal{E}([C])} \tag{6}$$

*where $e_c \in \mathbb{R}^C$ is the c-th one-hot codeword and $\|x\|_A \triangleq \sqrt{x^T A x}$ is referred as the Mahalanobis norm with positive definite matrix $A$.*

**Remark 1.** *From Proposition 1 we observe that, in NTK regime, ECOCs affect the performance of DNNs through the correlation matrix $\mathcal{E}([C])^T \mathcal{E}([C])$ of the codewords when the DNNs are free of weight-errors. For both one-hot and ECOCs, their target spaces $\mathbb{R}^{n_L}$ are equipped with $l_2$-norm and the corresponding decoding metrics. Proposition 1 suggests that ECOCs generalize the decoding metric. **Applying ECOCs is equivalent to equipping the target space of one-hot code with the Mahalanobis norm and the corresponding decoding metric.***

*Although it remains unclear which correlation matrices yield the best performance for ECOCs in absence of weight-errors, we know that codewords that are approximately orthogonal generally perform comparably to one-hot codes. Consequently, it is reasonable to regularize the orthogonality of the codewards during the ECOC construction.*

## 4.3 Efficacy of ECOCs on the Robustness of DNNs

It has been reported in multiple works that ECOCs can improve the robustness of DNNs against weight-errors [8, 9, 11]. However, these studies lack a rigorous theoretical explanation of why DNNs become more robust after ECOC application. In this section, we provide theoretical guarantee on the error correction capability of ECOCs. We begin with the weight-error model.

### 4.3.1 Weight-error Model

We consider the weight-errors proportional to the weight scale. Considering the typically small initial values of neural network weights and the minimal changes from these initial values after training in the NTK regime [14], we assume that the noise terms are independent and identically distributed (i.i.d.) Gaussian variables with zero mean and variance $\bar{\sigma}^2/n$.

### 4.3.2 Theoretical Guarantee for Robustness with ECOCs

We first show the bound of hidden layer outputs in absence of weight-errors, which will be used to prove other results.

**Lemma 1** (Hidden layer output bound). *Let Assumptions 1, 2, 3, and 4 hold. Then, for any hidden layer $l \leq L - 1$ and any $\delta > 0$,*

$$\frac{\|x_l\|_2}{\sqrt{n_l}} \leq \left(1 - \frac{B + |\phi(0)|}{1 - B}\right) B^l + \frac{B + |\phi(0)|}{1 - B} + \delta \tag{7}$$

*holds with probability at least $1 - \delta$ when $n$ is large enough, where the hidden layer width $n_l = \alpha_l n$ with constant $\alpha_l > 0$ for all $1 \leq l \leq L - 1$.*

Based on the lemma, we show the perturbation bound in the following theorem.

**Theorem 1** (Perturbation bound). *Let Assumptions 1, 2, 3, and 4 hold. Adopt the weight-error model in Sec. 4.3.1, and denote $\sigma^2 = \max_l \alpha_l \bar{\sigma}^2$. Let $x_L$ and $\tilde{x}_L$ denote any clean output (in absence of weight-errors) and its perturbed counterpart due to weight-errors, respectively. Then for arbitrary $\delta > 0$ and arbitrary DNN input, we have*

$$\frac{\|x_L - \tilde{x}_L\|_2}{\sqrt{n_L}} \leq \Xi(\sigma, B, |\phi(0)|, L) + \delta \tag{8}$$

*with probability at least $1 - \delta - o(n_L^{-1} \delta^{-1})$ when $n$ and $n_L$ are large enough, where*

$$\Xi(\sigma, B, |\phi(0)|, L) = \sigma B^L \left(1 - \frac{B + |\phi(0)|}{1 - B}\right) \frac{\sqrt{1+\sigma^2}^L - 1}{\sqrt{1+\sigma^2} - 1} + B\sigma \frac{1 + |\phi(0)|}{1 - B} \frac{\left(B\sqrt{1+\sigma^2}\right)^L - 1}{B\sqrt{1+\sigma^2} - 1}. \tag{9}$$

**Remark 2.** *The theorem suggests that the normalized output perturbation $\frac{\|x_L - \tilde{x}_L\|_2}{\sqrt{n_L}}$ can be bounded in terms of weight-error scale $\sigma$, Lipschitz constant $B$, absolute activation function value $|\phi(0)|$ and DNN depth $L$ with high probability when hidden layer width $n$ and code length $n_L$ are large enough. It can be observed that: 1) The bound can be made arbitrarily small when $\sigma = 0$, suggesting that the bound is tight. 2) The bound can be made arbitrarily small when $B = 0$, which means the activation function $\phi$ always outputs a constant regardless of its input. Obviously, in this situation, there will be no output perturbation, but training this DNN will be meaningless. 3) The DNN depth $L$ on the exponents accounts for the error propagation due to the feed-forward architecture of the DNNs.*

Based on Theorem 1, we present our main result in the following corollary to characterize the error correction capability of ECOCs.

**Corollary 1** (Main result). *Let all the conditions in Theorem 1 hold. Denote normalized ($l_2$) distance of a codeword $\mathcal{E}(i)$ and normalized uncertainty of clean prediction given the clean output $x_L$ as $dist(\mathcal{E}(i))$ and $U(x_L)$, respectively, with the following definition:*

$$dist(\mathcal{E}(i)) = \min_{j:j \neq i} \frac{1}{\sqrt{n_L}} \|\mathcal{E}(i) - \mathcal{E}(j)\|_2, \quad U(x_L) = \min_i \frac{1}{\sqrt{n_L}} \|\mathcal{E}(i) - x_L\|_2. \tag{10}$$

*Then a DNN with an ECOC can make prediction with $\tilde{x}_L$ as if it is free of weight-errors after decoding, i.e., $D(\tilde{x}_L) = D(x_L)$ with probability arbitrarily close to 1, if the ECOC satisfies*

$$\frac{dist(D(x_L))}{2} > U(x_L) + \Xi(\sigma, B, |\phi(0)|, L) + \delta \tag{11}$$

*for arbitrary small $\delta > 0$ when $n, n_L \to \infty$.*

**Remark 3.** *In eq. (40), $U(x_L)$ accounts for the distance between clean output $x_L$ and its closest codewords, and $\Xi(\sigma, B, |\phi(0)|, L) + \delta$ accounts for the output perturbation due to weight-errors. Note that Corollary 1 specifies conditions under which predictions can be made as if the DNN is free of weight-errors; however, it does not guarantee that these predictions will be the ground-truth labels. Even in the absence of weight-errors, DNNs may still produce incorrect predictions due to the data and model generalization, which cannot be corrected by ECOCs.*

A geometric explanation of Corollary 1 is given in Figure 1. The perturbed output $\tilde{x}_L$ (red diamond) is within the ball centered at clean output $x_L$ (blue dot) with radius $\Xi + \delta$ according to Theorem 1. If eq. (40) holds, then distance between $\tilde{x}_L$ and the codeword $D(x_L)$ (black dot at the bottom left), which is the decoding result of clean output $x_L$, will be smaller than the distance from $\tilde{x}_L$ to any other codewords (black dot at top right), mean that $D(x_L) = D(\tilde{x}_L)$ after decoding.

Figure 1: An illustration for Corollary 1.

## 5 ECOC Constructions

Inspired by our theoretical results, in this section, we propose two ECOC construction methods for small scale and large scale classification tasks, respectively.

### 5.1 Problem Formulation

The final performance of ECOCs is determined by weight-error free performance and performance degradation on top of that due to weight-errors, which are discussed in Proposition 1 and Corollary 1, respectively. According to Proposition 1, the weight-error free performance is influenced by the correlation matrices of ECOCs. Although the optimal correlation matrices remain unidentified, extensive experiments have verified that near orthogonal codewords are generally good. Therefore, we penalize the correlation of codewords during code construction. On the other hand, according to Corollary 1, larger normalized distance of codes leads to better error correction capability of ECOCs, therefore we encourage the distances of the codes during code construction. Let $Z \in \{-1, 1\}^{n_L \times C}$ be the ECOC codebook, i.e., a horizontal stack of codeword, then we construct the code by solving the following optimization problem:

$$\min_{Z \in \{-1,1\}^{n_L \times C}} -\underbrace{\sum_{i,j:i \neq j} \|Z[i] - Z[j]\|^2}_{\text{pair-wise distance}} + \lambda \underbrace{\left( \sum_{i \neq j} (Z[i]^T Z[j])^2 - \beta \sum_i \|Z[i]\|^2 \right)}_{\text{correlation}}. \tag{12}$$

Note the second part of the objective penalizes the magnitude of off-diagonal elements while promoting the amplitude of diagonal elements in the correlation matrix $Z^T Z$.

### 5.2 Method 1: Direct Optimization

Before employing standard optimization algorithms, it is necessary to relax the feasible set from the discrete binary domain $\{-1, 1\}^{n_L \times C}$ to the continuous interval $[-1, 1]^{n_L \times C}$. To further eliminate these box constraints, we reparameterize $Z$ with $Z = tanh(Z')$. This allows the application of gradient descent to effectively solve the optimization problem. Finally, we take the sign of each elements in $Z$ as the generated codebook.

### 5.3 Method 2: Picking from Hadamard

Since the objective in eq. (12) is non-convex, Method 1 can be easily trapped in local optima, especially for ECOCs with large number of classes $C$. For this reason, we introduce Method 2 in this section which constructs ECOCs on top of Hadamard codes. Hadamard codes exhibit several beneficial properties: 1) the number of codewords is equal to the code length, and both are powers of 2; 2) all codewords are orthogonal to each other; 3) the Hamming distance between any two codewords is half of the code length. This property allows Hadamard codes to achieve the upper bound of the minimum Hamming distance of ECOCs given the code length. Hadamard code is an excellent choice for ECOCs, which has been validated in previous works [10, 11, 12, 16]. An example of Hadamard code with code length $8$ is given in Section A.2.

In Method 2, we pick $C$ codewords from Hadamard codes and their complementary codes. Without loss of generality, we decompose $C$ into $C = 2a + b$, where $a$ and $b$ are non-negative integers. Let

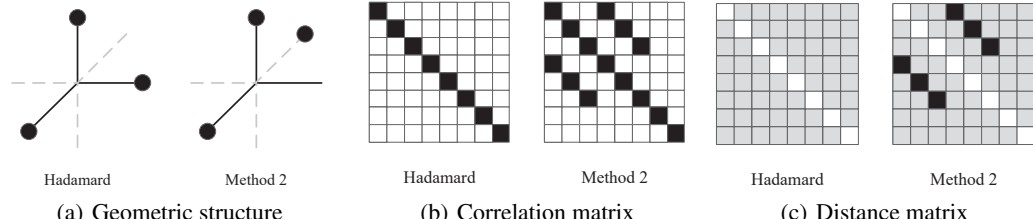

| Hadamard | Method 2 | Hadamard | Method 2 | Hadamard | Method 2 |

(a) Geometric structure     (b) Correlation matrix     (c) Distance matrix

Figure 2: A comparison between Hadamard codes and the codes constructed by Method 2. (a) Geometric structure: the codewords of Hadamard code are orthogonal to each other. In contrast, Method 2 allows one codeword to be located in the opposite direction from another. (b) Correlation matrix: correlation matrix for the codes constructed by Method 2 contain off-diagonal elements, indicating a higher level of correlation compared to that of Hadamard codes. (c) Distance matrix: codewords of Hadamard codes have uniform Hamming distances, whereas Method 2 produces some codeword pairs with larger Hamming distances and others are equivalent to those found in Hadamard codes, where the colors black, grey, and white are values of 8, 4, and 0, respectively.

$\{v_1, v_2, ..., v_H\}$ be the Hadamard codewords, then the code constructed by Method 2 is given by

$$\mathcal{E}_{a,b}^{pick}(i) = \begin{cases} v_i, \text{ for } 1 \leq i \leq a \\ -v_{i-a}, \text{ for } a < i \leq 2a \\ v_{i-a}, \text{ for } 2a < i \leq C. \end{cases} \tag{13}$$

In other words, its codebook $\mathcal{E}_{a,b}^{pick}([C]]) = \{v_1, v_2, ..., v_a, -v_1, -v_2, ..., -v_a, v_{a+1}, v_{a+2}, ..., v_{a+b}\}$. We can observe that for a codeword $\mathcal{E}_{a,b}^{pick}(i) = v_i$ with $i \leq a$, it is orthogonal to all other codewords except $\mathcal{E}_{a,b}^{pick}(i + a) = -v_i$. In addition, the Hamming distance between it and other codewords is half of the code length $\frac{n_L}{2}$, except $\mathcal{E}_{a,b}^{pick}(i + a)$, where the Hamming distance is $n_L$. When $a = 0$, codebook $\mathcal{E}_{a,b}^{pick}([C]]) = \{v_1, v_2, ..., v_C\}$ is composed of codewords from a Hadamard code.

Figure 2 provides a more intuitive comparison between Hadamard codes and the codes constructed by Method 2. Figure 2(a) illustrates the geometric structure of Hadamard codes and the codes constructed by Method 2 with $C = 3, a = 1$. As previously mentioned, the codewords of a Hadamard code are orthogonal to each other. In contrast, some pairs of codewords in Method 2 are located in the opposite direction from another. Figure 2(b) compares the correlation matrices of Hadamard code with code length $n_L = 8$ and the code constructed by Method 2 with $C = 8, a = 3$. We observe that the correlation matrix of Method 2 contains off-diagonal elements, indicating a higher level of correlation compared to that of Hadamard codes. Figure 2(c) shows the distance matrices of Hadamard code and Method 2 with $C = 8, a = 3$. We observe that the codewords of Hadamard codes have uniform Hamming distances (the number of different entries), whereas the codewords constructed by Method 2 produces some codeword pairs with larger Hamming distances and others that are equivalent to those found in Hadamard codes. Notice that, by adjusting $a$ and $b$ we can achieve different trade-off between averaged codeword distance and correlations.

## 6 Experiments

### 6.1 Experimental Setup

We conducted all simulations using PyTorch framework on a workstation equipped with an AMD EPYC 7542 32-Core Processor and four NVIDIA RTX A6000 GPUs, each with 40GB of memory.

**Datasets and models** MNIST [17], CIFAR10 and CIFAR100 [18], and Tiny ImageNet [19] datasets are used for evaluation. In our experiments, MNIST and CIFAR10 represent small scale tasks, while CIFAR100 [18] and Tiny ImageNet represent large scale tasks. We adopt a (784-4096-4096-10) multilayer perceptron (MLP), AlexNet, VGG16 and ResNet-50 for MNIST, CIFAR10, CIFAR100 and Tiny ImageNet, respectively.

**Benchmarks**  Our ECOCs are compared against one-hot codes, repetition codes, random codes, and Hadamard codes [10]. For a given number of classes, both the number of codewords and code length for one-hot codes are the same as the number of classes. We further derive a repetition code from a one-hot code: each codeword of a one-hot code is repeated for multiple times. Hence, the number of codewords in a repetition code is the same as the number of classes, but its code length is a multiple of the number of classes. The codewords of a random code is obtained by choosing random binary entries of $-1$ and $1$ with equal probability. For small scale tasks such as MNIST and CIFAR10, we compare codes constructed using Method 1 to with benchmarks. For large scale tasks, i.e., CIFAR100 and Tiny ImageNet, codes constructed based on Method 2 are used for comparison.

**Weight-errors**  Gaussian noises with different variance $\omega^2$ are added to the weights during inferences to test the robustness of the networks. For each code, except for random codes, the network is trained 20 times. After each training session, we perform 200 inference runs, each with different randomly generated weight-errors applied. For random codes, each training uses a different randomly generated codes.

## 6.2   Results and Analysis

**Superior performance of the proposed methods over existing ECOCs**  The performances of different ECOCs on small scale tasks (MNIST and CIFAR10) and large scale tasks (CIFAR100 and Tiny ImageNet) are summarized in Tables 1 and 2, respectively. We made the following observations from Tables 1 and 2: 1) The clean accuracy ($\omega = 0$) of different codes are similar for small scale tasks in Table 1, even with different code lengths. We believe this is partially due to the simplicity of the tasks. Besides, these codes all have good orthogonality, therefore they have similar generalization performance according to Proposition 1. In contrast, the clean accuracy is more varied for large scale tasks in Table 2. We believe this is because the assumption of infinitely wide DNN does not hold when the code length is comparable to the width of networks. In these cases, ECOCs with better structures can reduce more generalization error. 2) One-hot and repetition codes have much worse performance than other codes under large weight-errors. This is due to their smaller normalized distances. On the contrary, our proposed codes outperform other benchmarks under large weight-errors because of their larger normalized distances. 3) Larger code length $n_L$ yields better robustness under weight-errors since more protections are applied by the additional nodes. Besides, longer codes lead to higher success probability of bounds in our theoretical results, making the proposed methods inspired by them more effective. 4) Interestingly, Hadamard and random codes perform similarly when the code length is large. This is because the expectation of correlation matrices and pair-wise distances of

Table 1: Performance of different ECOCs with various code length ($n_L$) on small scale tasks, i.e., MNIST and CIFAR10. The proposed method achieves better performance in the presence of weight-errors.

| | $\omega$ | One-hot | Repetition | Random | Hadamard [10] | Method 1 |
|---|---|---|---|---|---|---|
| | | $n_L = 10$ | $n_L = 20$ | $n_L = 16$ | $n_L = 16$ | $n_L = 16$ |
| | 0 | $98.44 \pm 0.05$ | $98.45 \pm 0.05$ | $98.27 \pm 0.10$ | $98.48 \pm 0.04$ | $\mathbf{98.51} \pm 0.05$ |
| | 0.03 | $88.67 \pm 0.75$ | $89.60 \pm 0.62$ | $87.27 \pm 2.81$ | $\mathbf{93.54} \pm 0.18$ | $93.43 \pm 0.23$ |
| | 0.05 | $35.46 \pm 1.26$ | $46.39 \pm 0.88$ | $45.59 \pm 4.53$ | $\mathbf{60.49} \pm 0.86$ | $58.35 \pm 0.94$ |
| | 0.08 | $11.61 \pm 0.46$ | $17.39 \pm 0.28$ | $19.65 \pm 1.42$ | $\mathbf{23.65} \pm 0.43$ | $23.31 \pm 0.36$ |
| | 0.1 | $10.80 \pm 0.26$ | $13.71 \pm 0.34$ | $14.94 \pm 0.58$ | $\mathbf{16.59} \pm 0.25$ | $16.48 \pm 0.33$ |
| MNIST | | | $n_L = 130$ | $n_L = 128$ | $n_L = 128$ | $n_L = 128$ |
| | 0 | | $98.40 \pm 0.04$ | $98.49 \pm 0.04$ | $\mathbf{98.51} \pm 0.04$ | $98.51 \pm 0.05$ |
| | 0.03 | | $94.10 \pm 0.15$ | $96.72 \pm 0.22$ | $96.90 \pm 0.05$ | $\mathbf{97.04} \pm 0.06$ |
| | 0.05 | | $58.43 \pm 0.69$ | $80.78 \pm 2.53$ | $83.52 \pm 0.89$ | $\mathbf{84.33} \pm 0.72$ |
| | 0.08 | | $24.54 \pm 0.41$ | $38.34 \pm 2.05$ | $40.27 \pm 0.89$ | $\mathbf{42.47} \pm 0.59$ |
| | 0.1 | | $17.59 \pm 0.25$ | $24.44 \pm 0.93$ | $25.43 \pm 0.39$ | $\mathbf{26.54} \pm 0.39$ |
| | | $n_L = 10$ | $n_L = 20$ | $n_L = 16$ | $n_L = 16$ | $n_L = 16$ |
| | 0 | $82.48 \pm 0.20$ | $\mathbf{82.50} \pm 0.28$ | $81.56 \pm 0.24$ | $82.08 \pm 0.30$ | $82.08 \pm 0.33$ |
| | 0.01 | $78.83 \pm 0.13$ | $78.92 \pm 0.16$ | $80.41 \pm 0.23$ | $81.09 \pm 0.25$ | $\mathbf{81.18} \pm 0.23$ |
| | 0.02 | $60.40 \pm 0.58$ | $61.10 \pm 0.48$ | $71.38 \pm 1.99$ | $75.78 \pm 0.31$ | $\mathbf{76.21} \pm 0.31$ |
| | 0.03 | $11.15 \pm 0.17$ | $13.93 \pm 0.48$ | $32.52 \pm 5.51$ | $45.66 \pm 0.74$ | $\mathbf{50.18} \pm 0.84$ |
| | 0.04 | $10.00 \pm 0.19$ | $10.01 \pm 0.04$ | $14.47 \pm 1.39$ | $17.43 \pm 0.25$ | $\mathbf{19.58} \pm 0.25$ |
| CIFAR10 | | | $n_L = 130$ | $n_L = 128$ | $n_L = 128$ | $n_L = 128$ |
| | 0 | | $\mathbf{82.59} \pm 0.21$ | $82.14 \pm 0.28$ | $82.30 \pm 0.20$ | $82.30 \pm 0.29$ |
| | 0.01 | | $80.90 \pm 0.17$ | $81.14 \pm 0.19$ | $81.27 \pm 0.13$ | $\mathbf{81.28} \pm 0.24$ |
| | 0.02 | | $72.70 \pm 0.29$ | $76.58 \pm 0.23$ | $76.82 \pm 0.17$ | $\mathbf{77.15} \pm 0.24$ |
| | 0.03 | | $41.30 \pm 0.75$ | $54.45 \pm 1.93$ | $57.23 \pm 0.50$ | $\mathbf{59.52} \pm 0.65$ |
| | 0.04 | | $16.61 \pm 0.26$ | $21.08 \pm 0.92$ | $22.92 \pm 0.34$ | $\mathbf{24.52} \pm 0.47$ |

Table 2: Performance of different ECOCs with various code length ($n_L$) on large scale tasks, i.e., CIFAR100 and Tiny ImageNet. The proposed method shows better performance among all.

| | $\omega$ | One-hot | Repetition | Random | Hadamard [10] | Method 2 |
|---|---|---|---|---|---|---|
| | | $n_L = 100$ | $n_L = 1100$ | $n_L = 1024$ | $n_L = 1024$ | $n_L = 1024$ |
| | 0 | $62.81 \pm 0.03$ | $64.95 \pm 0.92$ | $67.37 \pm 0.35$ | $67.45 \pm 0.35$ | $\mathbf{68.25 \pm 0.22}$ |
| | 0.008 | $3.72 \pm 0.01$ | $29.83 \pm 5.03$ | $58.73 \pm 0.37$ | $60.41 \pm 0.32$ | $\mathbf{61.71 \pm 0.27}$ |
| | 0.01 | $1.56 \pm 0.00$ | $13.05 \pm 3.65$ | $49.08 \pm 0.84$ | $53.26 \pm 0.39$ | $\mathbf{55.69 \pm 0.54}$ |
| | 0.012 | $1.05 \pm 0.00$ | $4.78 \pm 1.53$ | $30.64 \pm 1.57$ | $38.87 \pm 0.95$ | $\mathbf{44.49 \pm 1.07}$ |
| | 0.013 | $1.00 \pm 0.00$ | $3.07 \pm 0.92$ | $19.75 \pm 1.66$ | $28.63 \pm 1.21$ | $\mathbf{36.13 \pm 1.41}$ |
| CIFAR100 | | | $n_L = 2100$ | $n_L = 2048$ | $n_L = 2048$ | $n_L = 2048$ |
| | 0 | | $66.50 \pm 0.46$ | $67.98 \pm 0.32$ | $67.90 \pm 0.34$ | $\mathbf{68.42 \pm 0.14}$ |
| | 0.008 | | $39.58 \pm 6.60$ | $64.77 \pm 0.23$ | $65.26 \pm 0.25$ | $\mathbf{65.79 \pm 0.12}$ |
| | 0.01 | | $21.53 \pm 6.14$ | $62.39 \pm 0.23$ | $63.47 \pm 0.25$ | $\mathbf{64.00 \pm 0.13}$ |
| | 0.012 | | $8.47 \pm 3.24$ | $58.51 \pm 0.32$ | $60.74 \pm 0.25$ | $\mathbf{61.38 \pm 0.20}$ |
| | 0.013 | | $5.10 \pm 2.00$ | $55.60 \pm 0.46$ | $58.82 \pm 0.27$ | $\mathbf{59.55 \pm 0.12}$ |
| | | $n_L = 200$ | $n_L = 1200$ | $n_L = 1024$ | $n_L = 1024$ | $n_L = 1024$ |
| | 0 | $48.74 \pm 0.43$ | $49.36 \pm 0.21$ | $53.34 \pm 0.47$ | $53.15 \pm 0.41$ | $\mathbf{56.42 \pm 0.42}$ |
| | 0.001 | $42.99 \pm 0.26$ | $46.63 \pm 0.20$ | $52.96 \pm 0.38$ | $52.92 \pm 0.42$ | $\mathbf{56.24 \pm 0.38}$ |
| | 0.003 | $13.41 \pm 0.45$ | $26.24 \pm 0.28$ | $50.28 \pm 0.27$ | $51.04 \pm 0.43$ | $\mathbf{54.53 \pm 0.31}$ |
| | 0.005 | $2.29 \pm 0.07$ | $6.46 \pm 0.06$ | $44.19 \pm 0.18$ | $46.87 \pm 0.45$ | $\mathbf{50.69 \pm 0.29}$ |
| | 0.008 | $0.55 \pm 0.02$ | $0.92 \pm 0.02$ | $25.91 \pm 0.48$ | $34.02 \pm 0.64$ | $\mathbf{38.17 \pm 0.43}$ |
| Tiny ImageNet | | | $n_L = 2200$ | $n_L = 2048$ | $n_L = 2048$ | $n_L = 2048$ |
| | 0 | | $49.83 \pm 0.21$ | $54.60 \pm 0.43$ | $55.53 \pm 0.15$ | $\mathbf{57.64 \pm 0.37}$ |
| | 0.001 | | $47.66 \pm 0.18$ | $54.52 \pm 0.45$ | $55.39 \pm 0.07$ | $\mathbf{57.56 \pm 0.33}$ |
| | 0.003 | | $31.23 \pm 0.17$ | $53.65 \pm 0.39$ | $54.33 \pm 0.11$ | $\mathbf{56.93 \pm 0.28}$ |
| | 0.005 | | $9.46 \pm 0.16$ | $51.78 \pm 0.35$ | $52.13 \pm 0.16$ | $\mathbf{55.54 \pm 0.27}$ |
| | 0.008 | | $1.06 \pm 0.06$ | $46.07 \pm 0.17$ | $46.62 \pm 0.34$ | $\mathbf{51.73 \pm 0.25}$ |

Table 3: Performance of Method 1 and Method 2 on CIFAR100, CIFAR80, CIFAR40, CIFAR20, and CIFAR10. The datasets are constructed by taking the corresponding number classes from CIFAR100. We choose VGG16 as the model and code length $n_L = 1024$ for all codes. Method 1 is better at small tasks, i.e., CIFAR10; while Method 2 is better at large tasks, i.e., CIFAR 20, 40, 60, 80 and 100.

| | $\omega$ | 0 | 0.008 | 0.01 | 0.012 | 0.013 |
|---|---|---|---|---|---|---|
| CIFAR100 | Method 1 | $67.65 \pm 0.21$ | $61.18 \pm 0.19$ | $55.00 \pm 0.17$ | $42.75 \pm 0.45$ | $33.83 \pm 0.76$ |
| | Method 2 | $68.25 \pm 0.22$ | $61.71 \pm 0.27$ | $55.69 \pm 0.54$ | $44.49 \pm 1.07$ | $36.13 \pm 1.41$ |
| CIFAR80 | Method 1 | $69.47 \pm 0.13$ | $63.19 \pm 0.18$ | $57.47 \pm 0.27$ | $46.34 \pm 0.68$ | $38.08 \pm 0.83$ |
| | Method 2 | $69.52 \pm 0.18$ | $63.22 \pm 0.12$ | $57.56 \pm 0.25$ | $46.56 \pm 0.37$ | $38.21 \pm 0.86$ |
| CIFAR40 | Method 1 | $75.78 \pm 0.14$ | $71.60 \pm 0.16$ | $68.12 \pm 0.24$ | $62.67 \pm 0.48$ | $58.50 \pm 0.74$ |
| | Method 2 | $76.01 \pm 0.29$ | $71.83 \pm 0.23$ | $68.33 \pm 0.28$ | $63.17 \pm 0.41$ | $59.57 \pm 0.63$ |
| CIFAR20 | Method 1 | $81.59 \pm 0.50$ | $77.18 \pm 0.28$ | $73.69 \pm 0.29$ | $67.72 \pm 0.46$ | $63.34 \pm 0.74$ |
| | Method 2 | $81.88 \pm 0.37$ | $77.53 \pm 0.25$ | $74.27 \pm 0.34$ | $69.02 \pm 0.53$ | $65.68 \pm 0.48$ |
| CIFAR10 | Method 1 | $88.30 \pm 0.33$ | $84.61 \pm 0.24$ | $80.90 \pm 0.44$ | $72.74 \pm 1.20$ | $66.35 \pm 1.93$ |
| | Method 2 | $88.58 \pm 0.21$ | $84.67 \pm 0.33$ | $80.85 \pm 0.54$ | $72.54 \pm 1.12$ | $66.19 \pm 1.27$ |

codeword in random codes are exactly same to those of Hadamard codes. When the code length is larger, the randomness of these quantities concentrate due to the law of large number.

**Performance comparison between Method 1 and Method 2** Recall that Method 1 and Method 2 are proposed for small scale and large scale tasks, respectively. Here, we compare the performance of Method 1 and Method 2 on tasks with different number of classes, which is summarized in Table 3. For small scale task, i.e., CIFAR10, Method 1 outperforms Method 2. However, for larger tasks with more classes, i.e., CIFAR20-100, Method 2 outperforms Method 1. As mentioned previously, we believe this is because Method 1 is more likely to be trapped in bad local minima in large scale tasks than small scale tasks. Additional experiments and analysis can be found in Section A.3.

## 7 Conclusion

In this paper, we provide theoretical explanations for the efficacy of ECOCs. We showed that using ECOCs to replace one-hot code is equivalent to changing the decoding metric from $l_2$ norm to a Mahalanobis norm corresponding to the correlation matrix of the codewords. Moreover, we provided an NN output perturbation bound, implying that the quantity normalized distance of codes is the key factor of the robustness of DNN with ECOCs. Based on the theoretical findings, we proposed two code construction methods for small and large scale classification tasks. Experimental results show that our proposed methods perform better than existing ECOCs.

## Acknowledgments and Disclosure of Funding

This work is partially supported by the National Science Foundation (NSF) under Grants No. 2349538 and No. 2401544.

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

## A    Supplementary Material

### A.1    Proofs for Theoretical Results

We start with a results in [14], which will be used in our proof.

**Theorem 2** (Theorem G.1 in [14]). *Let Assumptions 1, 2, 3, and 4 be satisfied. For any given $\delta > 0$, there exist constants $n' > 0$, $A > 0$, and $\eta_0 > 0$, such that for all $n > n'$ and any step t, employing gradient descent with a learning rate of $\eta = \frac{\eta_0}{n}$ ensures that*

$$\|\theta_t - \theta_0\|_2 \leq \frac{AR_0}{\sqrt{n}} \tag{14}$$

*with probability at least $1 - \delta$.*

#### A.1.1    Proofs of Proposition 1

**Proposition 1.** *Let Assumption 1, 2, 3 and 4 hold. Suppose there is no weight-errors and hidden layer width parameter $n \to \infty$, then applying ECOC is equivalent to changing the decoding function of one-hot codes (with 0 and 1 entries) to the following*

$$D(f(x)) = \arg \min_{c \in [C]} \|f(x) - e_c\|^2_{\mathcal{E}([C])^T \mathcal{E}([C])} \tag{15}$$

*where $e_c \in \mathbb{R}^C$ is the c-th one-hot codeword and $\mathcal{E}$ is the ECOC encoding function.*

*Proof.* Let $\tilde{\mathcal{Y}}$ denote the target encoded by the one-hot code. Notice that $\mathcal{E}(c) = \mathcal{E}([C])e_c$, then the ECOC target can be written as $\mathcal{Y} = \mathcal{E}([C])\tilde{\mathcal{Y}}$. The decoding process can be described as

$$\begin{aligned}
D(f(x)) &= \arg \min_{c \in [C]} \left\| \mathcal{E}(c) - \mathcal{Y}\mathcal{K}^{-1}(\mathcal{X}, \mathcal{X})\mathcal{K}(\mathcal{X}, x) \right\|^2 \\
&= \arg \min_{c \in [C]} \left\| \mathcal{E}([C])e_c - \mathcal{E}([C])\tilde{\mathcal{Y}}\bar{\mathcal{K}}^{-1}(\mathcal{X}, \mathcal{X})\mathcal{K}(\mathcal{X}, x) \right\|^2 \\
&= \arg \min_{c \in [C]} \left\| e_c - \tilde{\mathcal{Y}}\mathcal{K}^{-1}(\mathcal{X}, \mathcal{X})\mathcal{K}(\mathcal{X}, x) \right\|^2_{\mathcal{E}([C])^T \mathcal{E}([C])}.
\end{aligned} \tag{16}$$

$\square$

#### A.1.2    Proofs of Lemma 1

**Lemma 1** (Hidden layer output bound). *Let Assumptions 1, 2, 3, and 4 hold. Then, for any hidden layer $l \leq L - 1$ and any $\delta > 0$,*

$$\frac{\|x_l\|_2}{\sqrt{n_l}} \leq \left( 1 - \frac{B + |\phi(0)|}{1 - B} \right) B^l + \frac{B + |\phi(0)|}{1 - B} + \delta \tag{17}$$

*holds with probability at least $1 - \delta$ when $n$ is large enough, where the hidden later width $n_l = \alpha_l n$ with constant $\alpha_l > 0$ for all l.*

*Proof.* Consider $w_{ini,l}$ and $b_{ini,l}$ as the initial weights and bias of the $l$-th layer, respectively. Define $u_l$ and $v_l$ as the adjustments to the weights and bias of the $l$-th layer due to training, respectively, meaning $w_l = w_{ini,l} + u_l$ and $b_l = b_{ini,l} + v_l$. From eq. (3), we get

$$\begin{aligned}
h_l &= (w_{ini,l} + u_l)x_{l-1} + b_{ini,l} + v_l \\
&= w_{ini,l}x_{l-1} + b_{ini,l} + u_l x_{l-1} + v_l.
\end{aligned} \tag{18}$$

Applying the triangle inequality, we obtain

$$\|h_l\|_2 \leq \|w_{ini,l}x_{l-1} + b_{ini,l}\|_2 + \|u_l x_{l-1} + v_l\|_2. \tag{19}$$

Recall that the entries of $w_{ini,l}$ and $b_{ini,l}$ follow $\mathcal{N}(0, 1/n_{l-1})$ and $\mathcal{N}(0, 1)$, respectively, the elements of $(w_{ini,l}x_{l-1} + b_{ini,l})$ are independently and identically distributed Gaussian with zero mean and variance

$(\|x_{l-1}\|_2^2 / n_{l-1} + 1)$. The square of each element follows a $\chi^2$ distribution with mean $(\|x_{l-1}\|_2^2 / n_{l-1} + 1)$ and variance $2(\|x_{l-1}\|_2^2 / n_{l-1} + 1)$. Leveraging Chebyshev's inequality, for any $\delta_1 > 0$,

$$\frac{\|w_{ini,l}x_{l-1} + b_{ini,l}\|_2}{\sqrt{n_l}} \leq \sqrt{\frac{\|x_{l-1}\|_2^2}{n_{l-1}} + 1} + \delta_1 \leq \frac{\|x_{l-1}\|_2}{\sqrt{n_{l-1}}} + 1 + \delta_1 \tag{20}$$

with a probability of at least $1 - \frac{2(\|x_{l-1}\|_2^2/n_{l-1}+1)}{n_l\delta_1}$. For the second term,

$$\begin{aligned}
\frac{\|u_l x_{l-1} + v_l\|_2}{\sqrt{n_l}} &\leq \frac{\|u_l\|_F \|x_{l-1}\|_2}{\sqrt{n_l}} + \frac{\|v_l\|_2}{\sqrt{n_l}} \\
&= \frac{\sqrt{\alpha_{l-1}}}{\sqrt{\alpha_l}} \frac{\|u_l\|_F \|x_{l-1}\|_2}{\sqrt{n_{l-1}}} + \frac{\sqrt{\alpha_{l-1}}}{\sqrt{\alpha_l}} \frac{\|v_l\|_2}{\sqrt{n_{l-1}}} \\
&\leq \frac{AR_0}{\sqrt{n}} \frac{\sqrt{\alpha_{l-1}}}{\sqrt{\alpha_l}} \frac{\|x_{l-1}\|_2}{\sqrt{n_{l-1}}} + \frac{AR_0}{\sqrt{n}} \frac{\sqrt{\alpha_{l-1}}}{\sqrt{\alpha_l}} \frac{1}{\sqrt{n_{l-1}}} \\
&\leq \frac{AR_0}{\sqrt{n}} \frac{\sqrt{\alpha_{l-1}}}{\sqrt{\alpha_l}} \left( \frac{\|x_{l-1}\|_2 + 1}{\sqrt{n_{l-1}}} \right)
\end{aligned} \tag{21}$$

where the second inequality uses Theorem 2 and the definition of $\alpha_l = n_l/n$. Acknowledging Assumption 1, which posits $|\phi(h) - \phi(0)| \leq B|h|$, and combining eq. (19) with eqs. (20) and (21), we have

$$\begin{aligned}
\frac{\|x_l\|_2}{\sqrt{n_l}} &\leq \frac{B \|h_l\|_2}{\sqrt{n_l}} + |\phi(0)| \\
&\leq B \left( 1 + \frac{AR_0}{\sqrt{n}} \frac{\sqrt{\alpha_{l-1}}}{\sqrt{\alpha_l}} \right) \frac{\|x_{l-1}\|_2}{\sqrt{n_{l-1}}} + B + \frac{BAR_0}{\sqrt{n}} \frac{\sqrt{\alpha_{l-1}}}{\sqrt{\alpha_l}} \frac{1}{\sqrt{n_{l-1}}} + B\delta_1 + |\phi(0)|.
\end{aligned} \tag{22}$$

with a probability of at least $1 - \frac{2(\|x_{l-1}\|_2^2/n_{l-1}+1)}{n_l\delta_1}$. With sufficiently large $n$, we have

$$\frac{\|x_l\|_2}{\sqrt{n_l}} \leq \frac{B \|x_{l-1}\|_2}{\sqrt{n_{l-1}}} + (B + |\phi(0)|) + \delta' \tag{23}$$

with probability at least $1 - \delta'$ and $n$ large enough. After recursion, we have

$$\begin{aligned}
\frac{\|x_l\|_2}{\sqrt{n_l}} &\leq B^l \frac{\|x_0\|_2}{\sqrt{n_0}} + (B + |\phi(0)|) \frac{1 - B^l}{1 - B} + \delta \\
&\leq B^l + (B + |\phi(0)|) \frac{1 - B^l}{1 - B} + \delta
\end{aligned} \tag{24}$$

with probability at least $1 - \delta'$ and $n$ large enough, where the last inequality uses Assumption 2. $\qquad\square$

### A.1.3 Proofs of Theorem 1

**Theorem 1** (Perturbation bound). *Let Assumptions 1, 2, 3, and 4 hold. Adopt the weight-error model in Sec. 4.3.1, and denote $\sigma^2 = \max_l \alpha_l \bar{\sigma}^2$. Let $x_L$ and $\tilde{x}_L$ denote the clean output (in absence of weight-errors) and perturbed output due to weight-errors, respectively. Then for arbitrary $\delta > 0$ and arbitrary DNN input, we have*

$$\frac{\|x_L - \tilde{x}_L\|_2}{\sqrt{n_L}} \leq \Xi(\sigma, B, |\phi(0)|, L) + \delta \tag{25}$$

*with probability at least $1 - \delta - o(n_L^{-1}\delta^{-1})$ when $n$ and $n_L$ are large enough, where*

$$\Xi(\sigma, B, |\phi(0)|, L) = \sigma B^L \left( 1 - \frac{B + |\phi(0)|}{1 - B} \right) \frac{\sqrt{1 + \sigma^2}^L - 1}{\sqrt{1 + \sigma^2} - 1} + B\sigma \frac{1 + |\phi(0)|}{1 - B} \frac{\left(B\sqrt{1 + \sigma^2}\right)^L - 1}{B\sqrt{1 + \sigma^2} - 1}. \tag{26}$$

*Proof.* Let $\tilde{w}_{l+1}$ and $\tilde{b}_{l+1}$ denote the noisy weights and bias at the $(l+1)$-th layer, respectively, with the noise variances being $\sigma_w^2/n_l$ for the weights and $\sigma_b^2$ for the bias. Consequently, at the $l+1$-th layer, the relationships can be represented as:

$$\begin{cases} \tilde{h}_{l+1} = \tilde{w}_{l+1}\tilde{x}_l + \tilde{b}_{l+1}, \\ \tilde{x}_{l+1} = \phi(\tilde{h}_{l+1}). \end{cases} \tag{27}$$

Analyzing the preactivation term further, we obtain:

$$\begin{aligned}
\tilde{h}_{l+1} &= (w_{l+1} + \Delta w_{l+1})(x_l + \Delta x_l) + (b_{l+1} + \Delta b_{l+1}) \\
&= h_{l+1} + \Delta w_{l+1}x_l + (w_{l+1} + \Delta w_{l+1})\Delta x_l + \Delta b_{l+1} \\
&= h_{l+1} + \Delta w_{l+1}x_l + (w_{ini,l+1} + \Delta w_{l+1})\Delta x_l + u_{l+1}\Delta x_l + \Delta b_{l+1},
\end{aligned} \tag{28}$$

where $w_{ini,l+1}$ represents the initial weights at the $(l+1)$-th layer, characterized by a variance of $1/n_l$, and $u_{l+1}$ is the adjustment of weights as a result of the training process. The prefix $\Delta$ denotes the perturbation of corresponding variable due to the weight noise. Utilizing the triangle inequality, we deduce:

$$\left\|\tilde{h}_{l+1} - h_{l+1}\right\|_2 \leq \|\Delta w_{l+1} x_l\|_2 + \|(w_{ini,l+1} + \Delta w_{l+1})\Delta x_l\|_2 + \|u_{l+1}\Delta x_l\|_2 + \|\Delta b_{l+1}\|_2. \quad (29)$$

Recalling that the weight perturbation $\Delta w_{l+1}$ has a zero mean and a variance of $\sigma^2/n_l$ for $l \leq L-2$. Similar to eq. (20), Chebyshev's inequality gives

$$\frac{\|\Delta w_{l+1} x_l\|_2}{\sqrt{n_{l+1}}} \leq \frac{\sigma \|x_l\|_2}{\sqrt{n_l}} + \delta_1 \quad (30)$$

for any $\delta_1 > 0$, with the probability at least $1 - \frac{2\sigma^2\|x_l\|_2^2/n_l}{n_{l+1}\delta_1} \leq 1 - o(n^{-1}\delta_1^{-1})$.

Similarly, for the perturbed weights combined with the perturbed inputs, we obtain:

$$\frac{\|(w_{ini,l+1} + \Delta w_{l+1})\Delta x_l\|_2}{\sqrt{n_{l+1}}} \leq \frac{\sqrt{1+\sigma^2} \|\Delta x_l\|_2}{\sqrt{n_l}} + \delta_2 \quad (31)$$

for any $\delta_2 > 0$, with a corresponding probability of at least $1 - \frac{2(1+\sigma^2)\|\Delta x_l\|_2^2/n_l}{n_{l+1}\delta_2}$. For the bias perturbation, we have

$$\frac{\|\Delta b_{l+1}\|_2}{\sqrt{n_{l+1}}} \leq \sigma + \delta_3 \quad (32)$$

for any $\delta_3 > 0$ with the probability at least $1 - \frac{2\sigma_b^2}{n_{l+1}\delta_3} = 1 - o(n^{-1}\delta_3^{-1})$. According to Theorem 2, for arbitrary $\delta_4 > 0$, we have $\|u_{l+1}\|_F \leq AR_0 n^{-1/2}$ with probability at least $1 - \delta_4$ and $n$ large enough, which results in

$$\frac{\|u_{l+1}\Delta x_l\|_2}{\sqrt{n_{l+1}}} \leq \frac{\|u_{l+1}\|_F \|\Delta x_l\|_2}{\sqrt{n_{l+1}}} \leq \frac{AR_0}{\sqrt{n}} \frac{\sqrt{n_l}}{\sqrt{n_{l+1}}} \frac{\|\Delta x_l\|_2}{\sqrt{n_l}} = \frac{AR_0}{\sqrt{n}} \frac{\sqrt{\alpha_l}}{\sqrt{\alpha_{l+1}}} \frac{\|\Delta x_l\|_2}{\sqrt{n_l}}. \quad (33)$$

Note that this term can be arbitrarily small when $n$ is large enough. Combine eq. (29), (30), (31), (32) and (33), we have

$$\begin{aligned}
\frac{\|\Delta x_{l+1}\|_2}{\sqrt{n_{l+1}}} &\leq \frac{B\|\Delta h_{l+1}\|_2}{\sqrt{n_{1+1}}} \\
&\leq B\left(\frac{\sigma \|x_l\|_2}{\sqrt{n_l}} + \frac{\sqrt{1+\sigma^2} \|\Delta x_l\|_2}{\sqrt{n_l}} + \sigma\right) + \delta_l \\
&\leq B\left(\sigma\left(B^l + (B + |\phi(0)|)\frac{1-B^l}{1-B}\right) + \frac{\sqrt{1+\sigma^2} \|\Delta x_l\|_2}{\sqrt{n_l}} + \sigma\right) + \delta_l \\
&= B\sqrt{1+\sigma^2}\frac{\|\Delta x_l\|_2}{\sqrt{n_l}} + \sigma\left(1 - \frac{B + |\phi(0)|}{1-B}\right)B^{l+1} + B\sigma\frac{1 + |\phi(0)|}{1-B} + \delta_l
\end{aligned} \quad (34)$$

with probability at least $1 - \delta_l$ for $n$ large enough, where the first inequality uses Assumption 2, and the second inequality uses Lemma 1. Reuse eq. (34) for multiple times, we have for $l \leq L-1$

$$\begin{aligned}
\frac{\|\Delta x_l\|_2}{\sqrt{n_l}} &\leq \left(B\sqrt{1+\sigma^2}\right)^l \frac{\|\Delta x_0\|_2}{\sqrt{n_0}} + \sigma\left(1 - \frac{B + |\phi(0)|}{1-B}\right)\sum_{i=0}^{l-1}\left(B\sqrt{1+\sigma^2}\right)^i B^{l-i} \\
&\quad + B\sigma\frac{1 + |\phi(0)|}{1-B}\sum_{i=0}^{l-1}\left(B\sqrt{1+\sigma^2}\right)^i + \sum_{i=0}^{l-1}\left(B\sqrt{1+\sigma^2}\right)^i \delta_{n-i} \\
&= \sigma B^l\left(1 - \frac{B + |\phi(0)|}{1-B}\right)\sum_{i=0}^{l-1}\left(\sqrt{1+\sigma^2}\right)^i + B\sigma\frac{1 + |\phi(0)|}{1-B}\sum_{i=0}^{l-1}\left(B\sqrt{1+\sigma^2}\right)^i + \delta_l' \\
&= \sigma B^l\left(1 - \frac{B + |\phi(0)|}{1-B}\right)\frac{\sqrt{1+\sigma^2}^l - 1}{\sqrt{1+\sigma^2} - 1} + B\sigma\frac{1 + |\phi(0)|}{1-B}\frac{\left(B\sqrt{1+\sigma^2}\right)^l - 1}{B\sqrt{1+\sigma^2} - 1} + \delta_l'
\end{aligned} \quad (35)$$

with probability at least $1 - \delta_l'$ when $n$ is large enough, where the second inequality uses the fact $\|\Delta x_0\|_2 = 0$, and that $\delta_i$ can be arbitrarily small for all $i$ when $n$ is large enough.

We take care of the final layer specially due to the finite dimension. Specifically, we first rewrite eq. (33) as

$$\frac{\|u^L \Delta x_{L-1}\|_2}{\sqrt{n_L}} \leq \frac{AR_0}{\sqrt{n}} \frac{\sqrt{n_{L-1}}}{\sqrt{n_L}} \frac{\|\Delta x_{L-1}\|_2}{\sqrt{n_{L-1}}} \leq \frac{AR_0\sqrt{\alpha_{L-1}}}{\sqrt{n_L}} \frac{\|\Delta x_{L-1}\|_2}{\sqrt{n_{L-1}}}. \quad (36)$$

According to eq. (34), we have

$$\frac{\|\Delta h_L\|_2}{\sqrt{n_L}} \leq \frac{\sigma \|x_{L-1}\|_2}{\sqrt{n_{L-1}}} + \frac{\sqrt{1+\sigma^2}\|\Delta x_{L-1}\|_2}{\sqrt{n_{L-1}}} + \frac{AR_0\sqrt{\alpha_{L-1}}}{\sqrt{n_L}}\frac{\|\Delta x_{L-1}\|_2}{\sqrt{n_{L-1}}} + \sigma + \delta'_L \qquad (37)$$

for arbitrary $\delta'_L > 0$ with probability at least $1 - \delta'_L$ and $n$ large enough. Note that the probability bound in eq. (29), (30), (31), (32) and (33) all have order $o(n_L^{-1}\delta^{-1})$. Plug in eq. (35) and use Lemma 1, we finally have

$$\frac{\|\Delta x_L\|_2}{\sqrt{n_L}} \leq \left(\frac{AR_0\sqrt{\alpha_{L-1}}}{\sqrt{n_L}} + 1\right)\left(\sigma B^L\left(1 - \frac{B+|\phi(0)|}{1-B}\right)\frac{\sqrt{1+\sigma^2}^L - 1}{\sqrt{1+\sigma^2} - 1}\right.$$
$$\left. + B\sigma\frac{1+|\phi(0)|}{1-B}\frac{\left(B\sqrt{1+\sigma^2}\right)^L - 1}{B\sqrt{1+\sigma^2} - 1}\right) + \delta \qquad (38)$$

with probability at least $1 - \delta - o(n_L^{-1}\delta^{-1})$ when $n$ is large enough.

$\square$

### A.1.4   Proofs of Corollary 1

**Corollary 1** (Main result). *Let all the conditions in Theorem 1 hold. Denote normalized ($l_2$) distance of a codeword $\mathcal{E}(i)$ and normalized uncertainty of clean prediction given the clean output $x_L$ as $dist(\mathcal{E}(i))$ and $U(x_L)$, respectively, with the following definition:*

$$dist(\mathcal{E}(i)) = \min_{j:j\neq i}\frac{1}{\sqrt{n_L}}\|\mathcal{E}(i) - \mathcal{E}(j)\|_2, \quad U(x_L) = \min_i \frac{1}{\sqrt{n_L}}\|\mathcal{E}(i) - x_L\|_2. \qquad (39)$$

*Then a DNN with an ECOC can make prediction with $\tilde{x}_L$ as if it is free of weight-errors after decoding, i.e., $D(\tilde{x}_L) = D(x_L)$ with probability arbitrarily close to 1, if the ECOC satisfies*

$$\frac{dist(D(x_L))}{2} > U(x_L) + \Xi(\sigma, B, |\phi(0)|, L) + \delta \qquad (40)$$

*for arbitrary small $\delta > 0$ when $n, n_L \to \infty$.*

*Proof.* Our goal is to prove that for any codeword $\mathcal{E}(i) \neq D(x_L)$, the distance $\frac{\|\tilde{x}_L - D(x_L)\|_2}{\sqrt{n_L}} \leq \frac{\|\tilde{x}_L - \mathcal{E}(i)\|_2}{\sqrt{n_L}}$, which suggests that $\tilde{x}_L$ is mapped to $D(x_L)$ instead of any other codewords after decoding. By triangle inequality, we have

$$\frac{\|\tilde{x}_L - D(x_L)\|_2}{\sqrt{n_L}} \leq \frac{\|\tilde{x}_L - x_L\|_2}{\sqrt{n_L}} + U(x_L) \leq \Xi(\sigma, B, |\phi(0)|, L) + \delta + U(x_L) \leq \frac{dist(D(x_L))}{2} \qquad (41)$$

where the second and the third inequality use eq. (8) and (40), respectively. Given the definition of $dist(D(x_L))$, we have $\frac{\|\mathcal{E}(i) - D(x_L)\|_2}{\sqrt{n_L}} \geq dist(D(x_L))$. By triangle inequality again, we have

$$\frac{\|\mathcal{E}(i) - \tilde{x}_L\|_2}{\sqrt{n_L}} \geq \frac{\|\mathcal{E}(i) - D(x_L)\|_2}{\sqrt{n_L}} - \frac{\|\tilde{x}_L - D(x_L)\|_2}{\sqrt{n_L}} \geq \frac{dist(D(x_L))}{2} \qquad (42)$$

where the second inequality uses eq. (41). Comparing eqs. (41) and (42), we complete the proof. $\square$

### A.2   Hadamard Codes

Hadamard codes exhibit several beneficial properties: 1) the number of codewords is equal to the code length, and both are powers of 2; 2) all codewords are orthogonal to each other; 3) the Hamming distance between any two codewords is half of the code length. This property allows Hadamard codes to achieve the upper bound for the minimum Hamming distance given the code length. Hadamard code is an excellent choice for ECOC, which has been validated in previous works [11, 12]. An example of Hadamard code with code length 8 is as follows:

$$H_8 = \begin{bmatrix} 1 & 1 & 1 & 1 & 1 & 1 & 1 & 1 \\ 1 & -1 & 1 & -1 & 1 & -1 & 1 & -1 \\ 1 & 1 & -1 & -1 & 1 & 1 & -1 & -1 \\ 1 & -1 & -1 & 1 & 1 & -1 & -1 & 1 \\ 1 & 1 & 1 & 1 & -1 & -1 & -1 & -1 \\ 1 & -1 & 1 & -1 & -1 & 1 & -1 & 1 \\ 1 & 1 & -1 & -1 & -1 & -1 & 1 & 1 \\ 1 & -1 & -1 & 1 & -1 & 1 & 1 & -1 \end{bmatrix} \qquad (43)$$

Table 4: Performance of codes constructed by Method 1 and Method 2 on AlexNet CIFAR10. $n_L = 1024$ for both codes.

| $\omega$ | 0 | 0.01 | 0.02 | 0.03 | 0.04 |
|---|---|---|---|---|---|
| Method 1 | $82.34 \pm 0.23$ | $81.33 \pm 0.18$ | $77.54 \pm 0.21$ | $65.63 \pm 0.60$ | $34.89 \pm 0.88$ |
| Method 2 | $82.00 \pm 0.28$ | $81.01 \pm 0.19$ | $76.71 \pm 0.25$ | $63.53 \pm 0.62$ | $34.71 \pm 0.75$ |

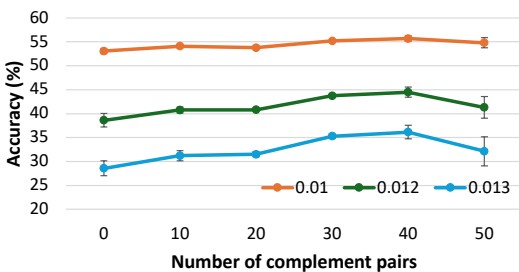

Figure 3: Accuracy comparisons for different number of complement pairs included when constructing ECOC with Method 2. Including more complement pairs means higher average distance while larger correlation. 0 complement pair is Hadamard code. In this plot, 40 complement pairs achieve the best trade-off between distance and correlation. The dataset, model and code length are CIFAR100, VGG-16 and 1024.

## A.3 Additional Experiments

**Performance comparison between Method 1 and Method 2**    Additionally, we compare the performance of ECOC constructed by Method 1 and Method 2 on small scale task with small scale DNN, i.e., AlexNet CIFAR10. Similarly, as shown in Table 4, Method 1 outperforms Method 2, which emphasizes the benefit of using Method 1 on small scale tasks.

**Performance comparison with correlation and distance trade-off**    In Section 4, we claimed that correlation of code matrices determines the weight-error free performance of ECOC and distance of codewords determines the performance degradation from weight-error. Here we use Method 2 as an example to show such trade-off. As shown in Figure 3, we vary the number of complement pairs $a$ in ECOC constructed by Method 2 and compare the performance. Larger $a$ means larger correlation while larger distance. In this set of experiments, $a = 40$ achieves the best correlation and distance trade-off.

## A.4 Limitation

In this paper, experimental results are from simulation without implementation on actual hardware accelerator. The theoretical results adopt assumptions of NTK, which requires the width of the network to approach to infinity. This assumption can be strong for some neural networks.

