# OpenReview forum: "Error Correction Output Codes for Robust Neural Networks against Weight-errors: A Neural Tangent Kernel Point of View"
_NeurIPS.cc/2024/Conference — NeurIPS 2024 poster_

### Official Review · Reviewer_vbnL · 2024-07-11

**Soundness:** 3
**Presentation:** 4
**Contribution:** 3
**Rating:** 7
**Confidence:** 3

**Summary:**

The authors applied ECOCs to DNNs to bolster the model’s resilience to weight errors. Most significantly, they have derived a perturbation bound through the utilization of the neural tangent kernel. According to their mathematical analysis, some principles of designing the ECOCs were revealed, leading to the construction of ECOCs based on Hadamard codes or direct optimization.  Experimental results indicate that the proposed method is effective.

I have not thoroughly examined the proofs of the theorems, but the theorems themselves appear to be plausible based on my understanding.

**Strengths:**

+ The perspective presented under the NTK is novel.
+ The mathematical analysis within this work is sound and offers valuable contributions to the academic community, particularly in the context of considering the one-hot code as a specialized form of ECOC.
+ The experimental results indicate that the proposed method exhibits a considerable performance advantage over existing approaches.
+ The manuscript is well written.

**Weaknesses:**

N/A

**Questions:**

+ Is there a parallel line of research analyzing the model’s robustness with one-hot/softmax outputs through the lens of the NTK?
+ Line 223, "Although the optimal correlation matrices remain unindentified". The authors should consider adding a relevant citation or providing an explanation to substantiate this claim. Could there be combinatorial approaches for constructing better codes?
+ Section 3.2. The use of $\ell$ (\ell in math mode) is preferred over $l$, as it helps to distinguish it from the numeral $1$.

---

> ### Author Rebuttal · Authors · 2024-08-05
>
> We are very thankful for your effort to provide constructive comments and support this work. Our responses to your comments are summarized below.
>
> ### **Response to questions**
> **Q1). Is there a parallel line of research analyzing the model’s robustness with one-hot/softmax outputs through the lens of the NTK?**
>
> A1). To the best knowledge of the authors, this is the first work analyzing ECOC's effectiveness on the robustness of NNs against weight-errors in the NTK regime. Although the softmax activation makes the learning dynamic more complicated, the perturbation of the final pre-activation caused by weight-errors can still be described by Theorem 1. With some analysis on the softmax function, the results could be extended. We thank the reviewer for pointing it out.
>
>
> **Q2). Line 223, "Although the optimal correlation matrices remain unindentified". The authors should consider adding a relevant citation or providing an explanation to substantiate this claim. Could there be combinatorial approaches for constructing better codes?**
>
> A2). We apologize for the confusion. Our key point is that the generalization performance (of the NTK model) depends on not only the correlation matrix but also the dataset (or the task). To identify the best correlation matrix, we need to know the data distribution (from which the dataset is sampled), which is often not available in real applications.
>
> The "combinatorial approaches" mentioned by the reviewer is actually a good point. While the searching space is very large considering the code length and number of classes, one possible direction could be to employ mixed integer programming. We will explore these approaches in our future work.
>
>
> **Q3). Section 3.2. The use of $\ell$  (\ell in math mode) is preferred over $l$ as it helps to distinguish it from the numeral $1$**
>
> A3). We thank the reviewer for the suggestion. We will make changes in the revised manuscript.

---

> > ### Comment · Reviewer_vbnL · 2024-08-13
> >
> > Thank you, I will keep my score.

---

### Official Review · Reviewer_Vn9W · 2024-07-12

**Soundness:** 3
**Presentation:** 2
**Contribution:** 2
**Rating:** 4
**Confidence:** 3

**Summary:**

The paper deals with the use of error-correcting output codes (ECOCs) for multi-class classification problem. The standard solution is to use one-hot code. Existing literature shows that there are codes which are better than one-hot code. At the same time there is a lack of theory and explanations of this phenomenon. The current paper aim to fill this gap and provide theoretical results which help to choose proper ECOC as well as provide guarantees. The authors utilize neural tangent kernel (NTK) approach for theoretical analysis.

**Strengths:**

The authors provide the results of theoretical analysis - this definitely the strong aspect of the paper. In particular, the main results are as follows:
- the authors showed that replacing one-hot code with other ECOC is equivalent to the change of decoding metric from Euclidean distance to Mahalanobis distance.
- the authors established an upper bound for the perturbation of DNN outputs (in case on weight-noise), which helps to choose the minimal distance of ECOC.
- two ECOC constructions are proposed: optimization-based construction and the construction based on Hadamard matrix.
- theoretical results are supported by numerical experiments.

**Weaknesses:**

I list the major weak points below:
1. The main contribution is repeated several times, namely in Introduction and in Section 4.
2. Please provide an explicit problem statement and describe the conditions. As I understand the main task is to deal with weight-error.
3. Please introduce the weight-error model explicitly. Now (in section 4.1) it is described by words but please define it formally. E.g. what does «	weight-errors proportional to the weight scale» mean? The result is w(1+z), where z ~ N(0, sigma^2) or it is not true?
4. If you have the weight-error model, then in my opinion, it would be better to train such noisy NN. In this case ECOC should be learned automatically. Could you please make such a comparison?
5. Corollary 1 seems to be trivial and follows directly from the coding theory. It says that (for the guaranteed recovery) the norm of the errors should be less or equal then d/2.
6. «Although it remains unclear which correlation matrices yield the best performance for ECOCs in absence of weight-errors, we know that codewords that are approximately orthogonal generally perform comparably to one-hot codes. Consequently, it is reasonable to regularize the orthogonality of the codewards during the ECOC construction». This is strange statement for the theoretically oriented paper. Actually the correlation is related to the minimum distance, namely $||a-b||^2 = ||a||^2 + ||b||^2 - 2 a^T b$ (codeword norms are usually fixed).
7. There are many ways to construct good codes over {+-1} alphabet. Usually you should start with good binary code (e.g. Reed-Muller code) and map 0 to +1 and 1 to -1. There exist tables of good binary codes. Please try to use them and compare to your solutions. I believe your Method 2 to be some particular case of this approach.
8. Experimental results demonstrate that constructed codes are better than one-hot code. At the same time my question is as follows. Is ECOC itself strategy beneficial? Could you please improve some state-of-the-art classifier with this approach?

**Questions:**

See Weaknesses

**Limitations:**

In my opinion, the limitation section should be increased, please discuss the limitations of your method rather that «hardware acceleration».

---

> ### Author Rebuttal · Authors · 2024-08-03
>
> Thank you for taking the time to review our paper. We appreciate your feedback and provide our responses below:
>
> **A1).** We will rephrase in the revised manuscript.
>
> **A2).** Recall that $f(\cdot; \theta)$ is the NN with weights $\theta$. Let $\mathcal{A} (\mathcal{E}, f, \mathcal{D};u)$ be a training algorithm, which takes the ECOC $\mathcal{E}$, NN architecture $f$, and dataset $\mathcal{D}$ as input and outputs the trained weights. Here $u$ accounts for the randomness in the algorithm. Define the test error $\mathcal{T}(\theta, \mathcal{D}) = \sum_{(x,c)\in \mathcal{D}}\mathbf{1}[D(f(x;\theta))-c]$, then the problem is
>
> $$\min_{\mathcal{E}} \mathsf{E}_{\Delta\theta} \mathsf{E}_u \mathcal{T}(\mathcal{A} (\mathcal{E}, f, \mathcal{D};u)+\Delta\theta, \mathcal{D})$$
> where $\Delta\theta$ is the random weight-errors. **Conditions:** NTK assumptions. This statement is for clarification, and we translate it into a trackable form in eq. (12).
>
>
> **A3).** Not True. The perturbed weights $\tilde{\theta} = \theta + \Delta\theta$, where entries of weight-errors $\Delta\theta$ are i.i.d. Gaussian with zero-mean and $\frac{\bar{\sigma}^2}{n}$-variance. We chose $\frac{\bar{\sigma}^2}{n}$ because: 1). On hardware devices, the noise is proportional to the scale of the weights. 2). In the NTK regime, the weights after training are of order $1/\sqrt{n}$ at the same scale of its initialization.
>
> **A4).** Noise-injection training (NIT) is an approach orthogonal to ECOC to combat weight-errors (see, e.g. [1]). However, it suffers from unstable training and long training time because each weight experiences variations independently in every training iteration. We compare the **accuracy** and **training time** of NIT and ECOC on MLP & MNIST task below, where vanilla is the original NN with one-hot code and NITs are noisy training NNs with different standard deviations.
>
> Accuracy:
> |method/noise level| 0 | 0.03 | 0.05 | 0.08 | 0.1 |
> |---------- |:----------: |  :---------: | :---------: | ---------: | :----------: |
> | vanilla | 98.44 | 88.67 | 35.46 | 11.61 | 10.80 |
> | NIT 0.03 | 98.51 | 98.02 | 93.44 | 24.56 | 11.42 |
> | NIT 0.05 | 98.16 | 97.85 | 96.66 | 73.25 | 23.50 |
> | NIT 0.1 | 10 | 10 | 10 | 10 | 10 |
> | ECOC (ours) | 98.51 | 97.04 | 84.33 | 42.47 | 26.54 |
> | ECOC (ours) + NIT 0.05 | 98.14 | 98.03 | 97.33 | 88.60 | 66.51|
>
> Training time (on NVIDIA RTX A6000 GPU)
> | Method | Training time |
> | ------------- | :----------: |
> | vanilla | 414s |
> | NIT | 3355s |
> | ECOC (ours) | 410s |
> | ECOC (ours) + NIT | 3309s |
>
> These results show that: 1) NIT improves the robustness but it significantly increases the training time; 2) training noise needs to be carefully picked when using NIT; 3) integrating the ECOC and NIT yields the best result.
>
> **A5).** Corollary 1 bears resemblance to the aforementioned coding theory result (i.e. half distance greater than the error), but is **different** in several aspects:
> 1) The definition of the minimum distance is different in that ours involves code length $n_L$ (see Eq. (10)).
> 2) The right hand side (RHS) of Eq. (11) is different from coding theory. In addition to weight-errors, which correspond to the
> coding theory result, the RHS of Eq. (11) accounts for the "confidence" of clean inference because the clean inference may fall far from a codeword.
> 3) While the classic coding theory result is deterministic, Corollary 1 is probabilistic.
>
> While Corollary 1 parallels the coding theory result, it **does not follow directly from the coding theory**. In fact, the derivation of Corollary 1 is quite involved and different from that for its counterpart in the coding theory. Hence, Corollary 1 is non-trivial.
>
> **A6).** We believe this stems from a misunderstanding of the our theoretical results. We claim that large codewords distance matters to the NN's robustness to weigh-errors. However, in terms of clean performance (in absence of weight errors), a good choice is to choose codewords to be orthogonal (based on the success of one-hot codes in absence of weight-errors), which does not directly relate to the codewords distance. In fact, our ECOC construction is a trade-off between clean performance and robustness, accounting for the final performance of NN with weight-errors.
>
> **A7).** Our Method 2 presented in Section 5.3 indeed constructs ECOC from good codes such as the Hadamard codes as suggested by the reviewer. Coincidentally, Method 2 uses a union of Hadamard codes and its complement, which is actually RM(1,N) code.
>
> Given a good binary code, the **challenge** is how to select a subset of codewords from it to form an ECOC. If codewords are randomly picked without methodology, it does not necessarily lead to the optimal performance. We compare the performances of these codes on the MNIST/MLP task:
> |method/noise level| 0 | 0.03 | 0.05 | 0.08 | 0.1 |
> | ------------- | :-----------: |  :----------: | :-----------: | :-----------: | :----------: |
> | Reed-Muller (n=128) | 98.51 | 96.81 | 81.52 | 39.27 | 24.98 |
> | Hadamard (n=128) | 98.51 | 96.90 | 83.52 | 40.27 | 25.43 |
> | Ours (n=128) | 98.51 | 97.04 | 84.33 | 42.47 | 26.54 |
>
> Our proposed method has better performance than ECOCs formed by randomly picking codewords from either the RM or Hadamard code.
>
> **A8).**  ECOC strategy itself can be beneficial only if codewords are properly designed ([2] shows a case of ECOC with lower clean accuracy than one-hot). Without loss of generality, we provide the results of ResNet-50/TinyImageNet suggesting ECOC is beneficial. We believe more followup researches should be conducted to further enrich the potentials of ECOC in NNs, which has been far less explored yet.
> |method| accuracy |
> | ------------- | :-----------: |
> | Vanilla | 48.74 |
> | Ours | 57.64 |
>
> [1] Y. Long et al. "Design of reliable DNN accelerator with un-reliable ReRAM." IEEE DATE, 2019.
>
> [2] A. Yu, et al. "COLA: orchestrating error coding and learning for robust neural network inference against hardware defects." ICML, 2023.

---

> > ### Author Response · Authors · 2024-08-12
> >
> > Dear Reviewer Vn9W,
> >
> > Thank you for reviewing our work and providing feedback that helps us improve the quality of our paper. We provided detailed responses according to your feedback, including some experiment results that you’re curious to see. In this regard, we hope you could take some time to take a look at our results. If you have any further questions or advice, please feel free to let us know. We’re looking forward to your kind response. We kindly remind that the deadline of the discussion period is tomorrow August 13th AOE. Your response is highly appreciated.
> >
> > Thank you!
> >
> > Regards,
> >
> > Authors

---

> > > ### Author Response · Authors · 2024-08-14
> > > **Gentle Reminder: Please kindly review our rebuttal (From the authors)**
> > >
> > > Dear Reviewer Vn9W,
> > >
> > > Thank you once again for your valuable feedback on our work. We wanted to kindly remind you that our rebuttal is now ready for your review. We fully appreciate your busy schedule and the additional effort this process requires.
> > >
> > > As the reviewer-author discussion period is concluding in just a few hours, and the other two reviewers have responded positively to our rebuttal, your response and advice will be crucial to our work. We would greatly appreciate your prompt attention to this matter.
> > >
> > > Thank you very much for your attention.
> > >
> > > Best,
> > >
> > > Authors

---

### Official Review · Reviewer_Nfve · 2024-07-13

**Soundness:** 3
**Presentation:** 3
**Contribution:** 4
**Rating:** 7
**Confidence:** 2

**Summary:**

This paper studies  theoretical foundations of Error correcting output code for multi-class classification by means of coding theory  and NTK.  NTK is employed to  alter the decoding metric from  l2 to Mahalanobis norm and  based on that a few  code construction methods are proposed.

**Strengths:**

The NTK  sees the model as an  Gaussian process, making it a  powerful tool for analysis of neural networks’ convergence and generalization. This paper  derives an interesting connection between decoding function in the infinite width regime  and NTK  formulation of Neural networks. The  derivation of bounds are  clear and  the codes are  tested  on different  tasks and datasets which is promising.

**Weaknesses:**

The presentation of the paper  could be further improved. Please consider the following:
- All parameters need to be defined in the main text. In  (6),  E([C])   and in (8)  $\bar{\delta}$ are not defined.
-There are sporadic grammatical errors such as
"... then training DNNs in the NTK regime minimizing MSE will result ...."
"... adopt a fully-connected feed-forward neural networks with L layers..."
"... can reduce more generalization error...."
"...each training uses a different randomly generated codes..."

- in eq (20), <= should be = .


-I see in the NTK papers a regularized form of  (4) as  $(K(x,x)+\lambda I)^{-1}$, therefore no need to   make assumption 3.
-NTK is merely used in the  proof of theorem 1 and is not the main  topic  of this paper.  I feel it is overemphasized in the title of the paper. please consider choosing a better title.

**Questions:**

-What is the reason  you present your work in the context of NTK?  I see that you make a  connection between ECOC  and NTK because both are in the infinite width regime. However you don't use NTK to analyze the training dynamics of the  networks.

- Given the sensitivity of your analysis  to width of the  networks, I expected some  figures demonstrating the  effect of increasing number of neurons on performance. Could you provide such a result to  show in what regime your codes are optimal?

**Limitations:**

NTK theory in general holds in the infinite width limit. Further discussions are  needed to adapt the methods in this paper to finite regime. For instance empirical NTK is developed in the literature to address the finite width regime.

---

> ### Author Rebuttal · Authors · 2024-08-02
>
> Thanks for taking your time to review and support our work. We appreciate your constructive feedback and provide our responses below:
>
> ### **Response to weakness 1**
> **Q1). All parameters need to be defined in the main text. In (6), $\mathcal{E}([C])$ and in (8) $\bar{\sigma}$
>  are not defined. -There are sporadic grammatical errors such as "... then training DNNs in the NTK regime minimizing MSE will result ...." "... adopt a fully-connected feed-forward neural networks with $L$ layers..." "... can reduce more generalization error...." "...each training uses a different randomly generated codes..."**
>
> **In eq (20), <= should be = .**
>
> A1) We apologize for the typos, grammar errors and missed definitions. We will fix all these issues.
>
> In Eq. (6), $\mathcal{E}([C])$ is an $n_L$-by-$C$ code matrix, where each column is a codeword $\mathcal{E}(i)$ for $1 \leq i \leq C$. In Eq. (8), $\bar{\sigma}$ is defined in Sec. 4.3.1 (weight-error model): we add noise with variance $\bar{\sigma}^2/n$ to the weights, where $n$ is a luxury variable defining the network width as specified in Sec. 3.2. Specifically, the $l$-th layer has a width of $\alpha_l n$ with a constant $\alpha_l$. The reasons behind this model are as follows: 1). On hardware devices, the noise is proportional to the scale of the weights. (The signal-to-noise ratio (SNR) matters.) 2). In NTK regime, the weights after training is of order $1/\sqrt{n}$ at the same scale of its initialization.
>
> For Eq. (20), we believe it should be $\leq$ because of Chebyshev’s inequality. Notice that in the proof, we only claim $n$ is large enough instead of approaching infinity. If $n$ approaches infinity, the first $\leq$ should be $=$ as suggested.
>
> ### **Response to weakness 2**
> **Q2). I see in the NTK papers a regularized form of (4) as $(\mathcal{K}(\mathcal{X},\mathcal{X})+I)^{-1}$, therefore no need to make assumption 3. -NTK is merely used in the proof of theorem 1 and is not the main topic of this paper. I feel it is overemphasized in the title of the paper. please consider choosing a better topic.**
>
> A2) We appreciate your valuable suggestions. First, we agree that Assumption 3 can be removed if $l_2$ regularization of the weights is applied during model training. Second, regarding the title, since the focus of this manuscript is to establish the foundation of ECOC design within the context of neural networks, with the NTK being used as a tool to analyze the behavior of neural networks trained with ECOC, we could modify the title as "On the Efficacy of Error Correction Output Codes for Robust Neural Networks Against Weight-Errors” per your suggestion if the conference rule allows.
>
> ### **Response to Questions**
> **Q3). What is the reason you present your work in the context of NTK? I see that you make a connection between ECOC and NTK because both are in the infinite width regime. However you don't use NTK to analyze the training dynamics of the networks.**
>
> A3). The reason we present the work in the context of NTK is as follows: Existing research on ECOC's applicability to modern deep neural networks (DNNs) mainly focuses on directly adopting well-known error correction codewords from the communication domain. It does not consider the unique information processing of DNNs, leading to suboptimal results. In our work, we want to provide guidance on ECOC design dedicated to DNNs through analyzing how DNNs behave after being trained with different codes. However, it is nontrivial given NN weight space is usually too complicated to analyze, we therefore leverage the NTK model to make the problem more tractable for the first time. In this regard, our focus is not on the dynamics of NTK itself, but rather on the consequences of training NNs (with ECOC) within the NTK regime.
>
> More specifically, our work relates to NTK from the following aspects:
> 1) In Proposition 1 (without weight-errors), we use the expression of NNs resulting from kernel-ridge regression (the dynamic of NTK).
> 2) Theorem 1 relies on the assumptions and intermediate results within the NTK regime.
>
>
>
> **Q4). Given the sensitivity of your analysis to width of the networks, I expected some figures demonstrating the effect of increasing number of neurons on performance. Could you provide such a result to show in what regime your codes are optimal?**
>
> A4) We added the experiment results on MNIST dataset with a 2-hidden layer MLP where the width of the layers is 512, 1024, 2048, or 4096. We construct the code using Method 1 with code length 128. The results are presented as follows:
>
> | width \ noise scale | 0 | 0.03 | 0.05 | 0.08 | 0.1 |
> | ----- | :------: | :-----: | :------: | :------: | :------: |
> | 512 | 98.43 | 89.75 | 59.82 | 26.86 | 18.67 |
> | 1024 | 98.48 | 93.23 | 66.52 | 28.80 | 19.37 |
> | 2048 | 98.50 | 95.48 | 73.40 | 32.30 | 20.99 |
> | 4096 | 98.51 | 97.04 | 84.33 | 42.47 | 26.54 |
>
> From the table we can observe that the robustness of NNs is improved when the network width $n$ becomes larger. This is because the success rate of the bound derived in the manuscript increases as $n$ becomes larger.
>
> ### **Response to limitations**
> In this work, only proposition 1 is based on the infinite width assumption. Both Theorem 1 and Corollary 1 assume $n$ is large enough instead of infinity (Our statement involves slack variable $\delta$ in the bound and the bound success rate in the form of ($1-...$)). We thank the reviewer for pointing out the empirical NTK. We leave refining the proposition 1 to the finite width version in future work.

---

> > ### Comment · Reviewer_Nfve · 2024-08-12
> >
> > Thank you for addressing my concerns,  I will keep my score.

---

### Decision · Program_Chairs · 2024-09-25

**Decision:**

Accept (poster)

**Comment:**

The authors have implemented Error-Correcting Output Codes (ECOCs) with Deep Neural Networks (DNNs) to enhance the model’s robustness against weight errors. Notably, they have derived a perturbation bound using the neural tangent kernel (NTK). Their mathematical analysis uncovers key design principles for ECOCs, leading to the development of ECOCs based on Hadamard codes or through direct optimization. Experimental results demonstrate the effectiveness of the proposed method. The NTK perspective introduced is novel and provides fresh insights.

The mathematical analysis presented is rigorous and makes a valuable contribution to the field, particularly in understanding the one-hot code as a specific type of ECOC. The experimental results show that the proposed method significantly outperforms existing approaches. Overall, the concept is compelling and offers notable advancements.